# A Dataset and Benchmark for 3D Part Recognition from 2D Images

## Abstract

While 3D semantic part understanding underpins numerous downstream applications, 3D part detection from images remains underexplored due to limited annotated datasets. To address this, we introduce DST-Part3D, a 3D semantic part dataset with $3,300$ fine-grained 3D part annotations across $475$ shapes from $50$ object categories, paired with $125,000$ realistic synthetic images. DST-Part3D enables training and evaluation of 3D part detection from images, 2D part segmentation via projection, and benchmarking of 3D correspondence quality through transferred part labels. Using this dataset, we develop Part321, an algorithm that recognizes 3D parts in images using only one annotated mesh per category. Part321 establishes mesh-to-mesh and mesh-to-image correspondences to propagate part pseudo-labels across instances, allowing effective 3D part detector training with minimal supervision. Experiments demonstrate that Part321 outperforms previous methods on 3D and 2D part detection tasks. In addition, we use DST-Part3D to analyze the mesh-to-mesh correspondence obtained by different methods leveraging transferred 3d part labels, highlighting the key challenge in 3D part correspondence, which provides insight into future work.

## 1 Introduction

Cognitive psychology studies suggest that humans recognize objects as a composition of simple geometric components in 3D space (Biederman, 1987). Many real-world vision applications, like robotics and autonomous driving, also require the ability to detect and segment object parts in 3D from single images. Specifically, fine-grained robot manipulation, such as lifting and rotating a bottle to display the label on the cap, requires robust reasoning about the object parts and their relationships with intended tasks (Yin et al., 2025). However, most existing works detect parts only in 2D since real-image datasets (Meletis et al., 2020; Zhou et al., 2017; Ramanathan et al., 2023; Chen et al., 2014; He et al., 2022; Liu et al., 2022) only have 2D part annotations for evaluation. In this paper, we introduce DST-Part3D, a 3D semantic part dataset paired with realistic synthetic images. We also develop Part321, an algorithm to recognize 3D parts in single images using only one annotated mesh, in order to flexibly detect parts under different part definitions with minimal supervision.

Annotating 3D parts on real images is nearly impossible without paired 3D object shapes. Therefore, we propose an alternative approach where we make a realistic synthetic dataset leveraging recent advances in incorporating 3D geometry control into diffusion models (Ma et al., 2024). We annotate $3,300$ 3D parts on $475$ shapes from $50$ object categories. Following 3D-DST (Ma et al., 2024), we leverage latent diffusion models with text prompts (Rombach et al., 2022), coupled with Control-Net (Zhang et al., 2023a), to generate realistic synthetic images conditioned on edge maps of rendered images. Using this method, we create a large-scale realistic synthetic dataset of $125,000$ images that have 3D part annotations. We name it DST-Part3D. DST-Part3D can not only enable training and evaluation of 3D part detection from images and 2D part segmentation via projection, but also enable benchmarking of 3D correspondence quality through transferred part labels.

Based on the proposed dataset, we study the problem of recognizing 3D parts from single images using one annotated mesh. We introduce Part321, where 3 denotes 3D part information, 2 denotes a single 2D image, and 1 denotes one annotated mesh. We need to address two challenges: (1) The geometry of different objects in one object category can vary significantly, *e.g.*, different shapes of

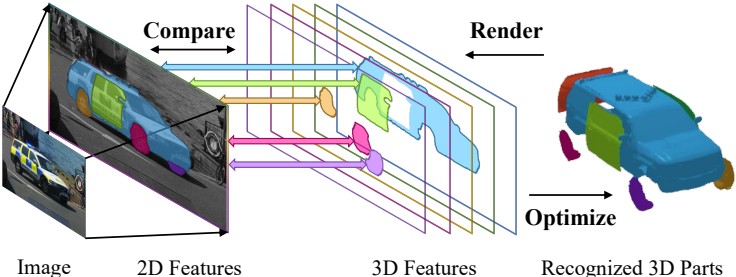

Figure 1: **Part321.** Our framework detects 3D object parts from a single image by extracting 2D features from the image and compare with rendered 3D features in deformable neural mesh. Through optimizing the 3D pose, scale and shape of object parts, we recognize them in 3D space.

airplanes. Thus, finding a commonly shared representation of diverse shapes is essential. (2) We need to model the semantic relation between the 3D parts and the 2D image pixels.

To address the first challenge, we establish mesh-to-mesh correspondence by using the one-shot annotated mesh as a template and matching each of its vertices to the most semantically similar vertices on other meshes within the same category. Consequently, we can share the vertex feature of any vertex on the annotated mesh with the corresponding vertex on other shape instances in the category. Using the obtained correspondences, we also train a deformation network that predicts vertex offsets conditioned on one-hot shape latents representing other meshes, enabling the template to deform into diverse geometries.

To solve the second challenge, we attach neural features (Wang et al., 2020; Shtedritski et al., 2024; Neverova et al., 2020) to the vertices of the deformable mesh, forming a deformable neural mesh. The neural features can be shared among different instances based on the mesh-to-mesh correspondence. Then we can learn the mesh-to-image correspondence, which aligns the neural features on the meshes with the 2D features extracted from images (Wang et al., 2020). Leveraging the camera poses and 3D shapes from DST-Part3D, we use contrastive learning to optimize the neural features to align with the 2D features extracted from corresponding pixels. Benefiting from the realism of our realistic synthetic images, this correspondence generalizes to real data as well.

During inference, as shown in Figure 1. Given a test image, we first extract the 2D features from the image and render 3D features from the deformable neural mesh. Through comparing the features and taking the derivative, we find clues to change the 3D pose, position, and shape of the whole neural mesh as well as each part to better align the two feature sets, leading to the optimized configuration of parts corresponding to the input image.

We evaluate Part321's pioneering ability of one-shot 3D part detection from single images on the DST-Part3D, which outperforms crafted baselines. To further validate the generalization ability of the model on real images, we compare Part321 with 2D segmentation methods on three real-world image part datasets by projecting the predicted 3D parts into the image plane. To analyze the performance bottleneck of Part321, we evaluate the mesh-to-mesh correspondence calculated by different methods using the proxy task of part transfer, which reveals the fundamental challenge of 3D part correspondence and provides insight for future improvement.

Our contributions can be summarized as follows:

- We construct DST-Part3D dataset, with 3D annotated parts for 50 object categories, 475 shapes, and $\sim 3,300$ parts, paired with $125,000$ realistic synthetic images.

- We propose Part321, a category-level object 3D part recognition method that only requires one annotated mesh, which pioneers one-shot 3D part detection from single images by learning two types of correspondence.

- Part321 outperforms designed baselines on one-shot 3D part detection from single images. It also outperforms 2D segmentation methods on real image part datasets.

- We use DST-Part3D to analyze the mesh-to-mesh correspondence obtained by different methods using part transfer, highlighting the challenge in 3D part correspondence.

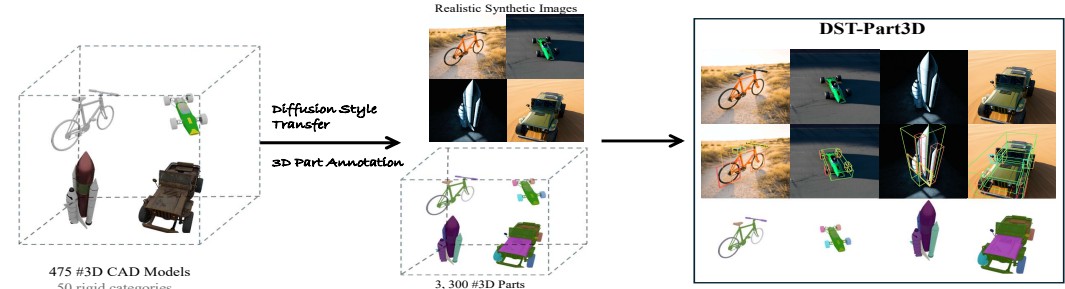

Figure 2: **Overview of the DST-Part3D Dataset**. Firstly, we select 475 3D CAD models from 50 rigid object categories. Secondly, we generate realistic synthetic images following the 3D-DST generation pipeline and annotate 3d parts on these meshes. Then we obtain DST-Part3D that has realistic synthetic images with 3D part annotations.

## 2 RELATED WORK

**3D Part Dataset.** Compared to 2D part datasets, 3D part datasets are rather limited. We have ShapeNet-Part (Yi et al., 2016) that annotates 16 rigid categories of 31.9k shapes, but the part definitions are coarse, averaging only about three parts per instance. PartNet (Mo et al., 2019) annotates 24 indoor object categories of 26.7k shapes with fine-grained hierarchical 3D part annotations. 3DCoM-PaT++ (Slim et al., 2023) annotates 10k shapes of 41 categories with even more fine-grained parts. While these datasets are large-scale, they are primarily designed for 3D part segmentation with 3D input. For instance, fine-grained datasets like PartNet and 3DCoMPaT++ include very small parts that are difficult to detect in 2D images. Moreover, only 3DCoMPaT++ provides rendered images of annotated meshes, but those synthetic images use only a white background. In contrast, we introduce DST-Part3D, a 3D part dataset paired with large-scale realistic synthetic images, which is specifically designed for recognizing 3D parts from single images.

**Learning Object Parts in 3D.** Learning 3D parts from 3D inputs, *e.g.*, pointclouds, has been widely explored. Previous works on 3D semantic segmentation have explored many effective network architectures (Qi et al., 2017a;b; Yu et al., 2019; Shi et al., 2020; Zhang et al., 2022) and training methods (Landrieu & Simonovsky, 2018; Afham et al., 2022; Liu et al., 2023a; Zhang et al., 2023b) to improve the ability of 3D part segmentation. Another important area is 3D part discovery, which involves decomposing 3D objects into self-defined parts (Xu et al., 2019; Luo et al., 2020; Sun et al., 2021; Koo et al., 2022). However, one limitation of these approaches is their reliance on 3D observations. In order to recognize 3D geometries from images, previous methods try to register pixels to a canonical mesh Kulkarni et al. (2020); Yang et al. (2021); Kokkinos & Kokkinos (2021). These methods focus on deformable articulated categories (*i.e.*, animals) where the articulation is hard but the geometric variance within a category is smaller than rigid objects. Animals have relatively fixed part configurations, but some part positions can change a lot across instances for rigid objects. More recent approaches focus on using multiview 2D part correspondence for 3D parts (Sharma et al., 2022; Thai et al., 2024), but they still suffer from low-quality 2D part segmentation under challenging situations (e.g., fine-grained part definitions, extreme poses, etc). In contrast, we leverage mesh-to-image correspondence to relax the requirement for 3D input and use mesh-to-mesh correspondence to overcome geometric variance across rigid object instances.

## 3 DATASET CONSTRUCTION

DST-Part3D aims to provide fine-grained 3D semantic part annotations for 50 common rigid object categories in the real world. The 50 categories - 40 outdoor and 10 indoor objects - are selected from 3D-DST (Ma et al., 2024) which classifies CAD models obtained from ShapeNet (Chang et al., 2015), Objaverse (Deitke et al., 2023), OmniObject3D (Wu et al., 2023), and Toys4k (Stojanov et al., 2021) into ImageNet-1k (Deng et al., 2009) categories. These fine-grained object categories can not only help us constrain the shape variance within each category but also benefit the generated prompts for latent diffusion models by including more shape-specific information (*e.g.*, describes one vehicle shape as a convertible rather than a general car). There are 761 shapes in these 50 categories, and we annotated parts on 475 of them — about 10 per category. Compared to hierarchical parts directly

from assets, our annotations emphasize semantic parts that align better with images by defining surface parts visible from typical viewpoints. On average, each annotated mesh contains 7 parts, yielding roughly $3,300$ parts in total. We also introduce spatial information to our part definitions(*e.g.* left door) due to the importance of accurate spatial recognition of 3D part configurations, and we merge tiny, hard-to-recognize parts into larger, semantically coherent parts. Additionally, following the generation pipeline in 3D-DST, we leverage a more advanced latent diffusion model (*i.e.* Stable Diffusion 3) to generate $125,000$ realistic synthetic images with 3D part annotations as shown in Figure 2. Note that we also generate realistic synthetic images on the unannotated meshes for training Part321. Please refer to the Appendix A.3 for more details about the dataset and experiments, which demonstrate the realism of the generated images. Furthermore, benefiting from our consistent semantic part definitions and constrained shape-variance within each fine-grained category, DST-Part3D naturally becomes a promising 3D part dataset to benchmark the quality of 3D correspondence by testing the transferred 3D part labels using 3D correspondence between two meshes.

## 4  PART321

We formulate the one-shot 3D object part recognition problem as establishing a deformable mesh attached with neural features (Section 4.1), which is a shared 3D representation among different 3D shapes in one object category and is aligned with 2D images. As shown in Figure 3, we build mesh-to-mesh correspondence (Section 4.2) and mesh-to-image correspondence (Section 4.3) to achieve this. Such a formulation makes detecting 3D parts from 2D images approachable by differentiating the process of rendering the deformable neural mesh into 2D feature maps.

During inference (Section 4.4), we first recognize the object using the mesh template without deformation to have a preliminary estimate of its 3D pose and part positions. Then we refine the 3D pose and part configurations by allowing deformation.

### 4.1  DEFORMABLE NEURAL MESH

We aim to establish a shared representation of diverse shapes in each object category. We implement it by using the mesh-to-mesh correspondence so that the vertex feature of the annotated one mesh can be shared among different shape instances. We also enable deformation for the annotated mesh by training a deformation network to predict vertex offsets. The deformation is based on the mesh-to-mesh correspondence and is conditioned on one-hot shape latents that represent other meshes. Note that the shape latents are independent for each vertex by default. Additionally, we attach neural features to the vertices of our deformable mesh model so that we can align the neural features on the meshes with the 2D features extracted from images by the mesh-to-image correspondence.

We name it deformable neural mesh: $\mathfrak{N} = \{\mathcal{V}, \mathcal{A}, \Theta, \mathcal{P}, \mathcal{Z}\}$, which consists mesh vertices $\mathcal{V} = \{V_k \in \mathbb{R}^3\}_{k=1}^{K}$, triangular faces $\mathcal{A} = \{A_k \in \mathbb{N}^3\}_{k=1}^{K'}$, feature vector on each vertex $\Theta = \{\theta_k \in \mathbb{R}^d\}_{k=1}^{K}$, part label $\mathcal{P} = \{P_k \in \mathbb{N}\}_{k=1}^{K}$, and one-hot shape latent $\mathcal{Z} = \{z_k \in \mathbb{R}^{d'}\}_{k=1}^{K}$, conditioned on which the mesh will be reshaped into diverse geometries. $K$ and $K'$ are the number of vertices and faces of the mesh.

### 4.2  MESH-TO-MESH CORRESPONDENCE

We use a set of meshes from the category $\{\mathfrak{M}_y = \{\mathcal{V}_y, \mathcal{A}_y\}\}$ to learn the vertex-level mesh-to-mesh correspondence, where $y$ is the index of the mesh and $\mathcal{V}_y$ and $\mathcal{A}_y$ are the vertices and faces of the $y$-th mesh.

We formulate the mesh-to-mesh correspondence as feature matching, which means that we propose to learn the features on each vertex representing its geometric information and exploit the cosine similarity of feature vectors to form the correspondence. We use a PointNet++ (Qi et al., 2017b) encoder $\Psi$ pre-trained on reconstruction (Sun et al., 2021) to extract object geometry descriptors from point clouds. It is self-supervisedly trained on $31,747$ shapes of 13 object categories collected from ShapeNet (Chang et al., 2015). We use it to compute vertex features on each mesh $\gamma_y = \Psi(\mathcal{V}_y)$. The features of each vertex $\mathcal{V}_{y,k}$ is $\gamma_{y,k}$, where $k$ is the index of vertex in mesh $\mathfrak{M}_y$. Then, we compute the cosine similarity of features to obtain the dense correspondence between vertices across all object

Figure 3: **Overview of the training process of Part321.** The mesh-to-mesh correspondence is learned between mesh vertices, which finds corresponding vertices across different objects in the category. Then, realistic synthetic images are generated using the 3D-DST (Ma et al., 2024), based on which the mesh-to-image correspondence is learned to align the 3D meshes with the generated 2D image using contrastive learning.

meshes. For example, for the $k_1$-th vertex on mesh $\mathfrak{M}_{y_1}$, the corresponding vertex on mesh $\mathfrak{M}_{y_2}$ is defined as:

$$Corr(y_1, k_1, y_2) = \operatorname{argmax}_k \frac{\gamma_{y_1, k_1} \cdot \gamma_{y_2, k}}{\|\gamma_{y_1, k_1}\| \|\gamma_{y_2, k}\|}. \tag{1}$$

With mesh-to-mesh correspondence, the corresponding vertices from different meshes could share the same feature vector in the annotated mesh. To make the annotated mesh deformable, we train a deformation network $\varphi$ based on the correspondence. We aim to make each part separately deformable to represent diverse combinations, but we also need the deformable mesh to be part-agnostic to avoid retraining when the part definition changes. Therefore, we attach one shape latent to each vertex in the annotated mesh. When we have the part definition during inference, we keep the shape latent of the vertices in the same part to be the same. We build an MLP that takes one vertex of the annotated mesh $V_k$, the shape latent $z_k$ attached to this vertex, and outputs the offset between the target position specified by the shape latent and the original position of $V_k$. Thus, the deformation is defined as:

$$\hat{V}_k = V_k + \varphi(V_k, z_k), k = 1, 2, ..., K \tag{2}$$

where $\hat{V}_k$ is the vertex after deformation. For training, the deformation network takes the vertex $V_k$ and a one-hot shape latent $h_y$ that represents the target mesh $\mathfrak{M}_y$. We use a different notation for shape latent here (*i.e.*, $h_y$) because during the training stage, the shape latents of all vertices are the same (meaning they deform to the $y$-th mesh). Based on mesh-to-mesh correspondence, the network should output the position offset between the corresponding vertex $\mathcal{V}_{y, Corr(y_0, k, y)}$ and $V_k$, where $y_0$ is the index of the annotated mesh in the category. The loss for training $\varphi$ is:

$$\mathcal{L}_{\text{deform}} = \sum_y \sum_k |(\mathcal{V}_{y, Corr(y_0, k, y)} - V_k) - \varphi(V_k, h_y)|. \tag{3}$$

We also apply the surface-normal consistency loss to keep the deformed mesh smooth. During inference, our framework could deform each object part into diverse shapes by changing the latent $\mathcal{Z}$. Please refer to the Appendix for more details on the deformation network.

### 4.3 MESH-TO-IMAGE CORRESPONDENCE

To relate the vertices in the deformable mesh with 2D images, we add neural features to its vertices and introduce the mesh-to-image correspondence, which is formulated as the similarity between features on each vertex $\theta_k$ and the features extracted $\Phi(I) = F \in \mathbb{R}^{c \times h \times w}$ from image $I$, where $\Phi$ is the feature extractor we need to train.

As shown in Figure 3, we use the realistic synthetic images in DST-Part3D generated by the meshes set $\{\mathfrak{M}_y\}$ for training. We only need to use the ground truth 3D pose and shape as the supervision, which requires no 3D part annotations.

We employ the same learning process to learn the correspondence as previous object pose estimation approaches using contrastive learning (Ma et al., 2022; Wang et al., 2020; 2024). However, rather

Figure 4: **The inference process of Part321.** We use the deformation network to reshape each part given the one-hot shape latent (represented by grids with grayscale values). Image features are extracted from the given test image. We optimize the whole object pose and 3D configuration (location, rotation, scale, and shape) of object parts by gradient-based minimizing the feature reconstruction loss. The 2D part segmentation is computed by a projection of optimized 3D parts.

than learning the vertex features fuzzily with a prototype geometry (*e.g.*, a cuboid), we utilize the detailed object geometry (*i.e.*, $\{\mathfrak{M}_y\}$) since we have the 3D correspondence to share the sampled vertex features across the meshes. Relying on detailed shapes makes it feasible to learn precise features $\theta_k$ of part-level structures, *e.g.*, the center of the left front wheel. Such a difference allows Part321 to locate object parts accurately. For details about the learning formulation, please refer to the Appendix A.1.1.

## 4.4 PART INFERENCE

Our inference pipeline (Figure 4) predicts 3D object parts by optimizing the overall object pose and 3D configurations (*i.e.*, location, rotation, scale, and shape) of each part in the deformable neural mesh via the feature-level rendering and comparison.

Specifically, we extract a feature map using the trained feature extractor from the testing image $F = \Phi(I)$. We also render a feature map $\hat{F}$ using the built neural mesh $\mathfrak{N}$ given a set of 3D configurations, *e.g.*, shape, pose, and scale. By comparing the two feature maps we update the 3D attributes to make the rendered feature map better align with the extracted features. Technically, we use gradient optimization to iteratively minimize the feature difference loss on each pixel $p$ on the feature map:

$$\mathcal{L}_{\text{recon}} = \sum_p \|F_p - \hat{F}_p\|^2. \tag{4}$$

In detail, we first optimize the camera pose $R$ by disabling deformation of the deformable neural mesh. We start by sampling 144 initial poses and choose the one with the smallest feature difference loss for further optimization, which gives a preliminary estimate of the 3D rotation $R$ of the whole object and its part positions. Secondly, we refine the 3D pose and part configurations by allowing deformation. For each object part, we optimize the scaling parameter $S \in \mathbb{R}^3$, shape latent $\mathcal{Z}$, and transformation $T \in \mathbb{R}^6$, which includes the 3D translation and 3D rotation. By changing $T$, $S$, and $\mathcal{Z}$, each object part can move freely in 3D space and be deformed into diverse shapes.

Also, to ensure the geometry consistency between object parts, we introduce a geometry consistency loss. We select all the paired vertices $\{V_k, V_l\}, P_k \neq P_l$ that belong to different object parts, which have distances $\rho_{kl} = \|V_k - V_l\|$ smaller than a threshold $\tau$. A consistency loss is applied if the distance of these paired vertices exceeds the threshold during optimization:

$$\mathcal{L}_{\text{consist}} = \sum_{k,l} (\rho_{kl} - \tau)\mathbb{1}[\rho_{ij} > \tau], \tag{5}$$

where $\mathbb{1}$ is an indicator function that equals 1 if the expression is true and equals 0 if otherwise.

The overall optimization loss $\mathcal{L}_{\text{inference}}$ is the weighted sum of feature difference loss and geometry consistency loss with $w_{\text{consist}}$ as the weight. We conduct gradient optimization to find 3D configuration w.r.t. $R, T, S, \mathcal{Z}$ with minimal $\mathcal{L}_{\text{inference}}$ for 300 steps, thus recognizing object parts in 3D space.

| | | | | | | | | | | | | | | | | | | | | | | | | |
|---|---|---|---|---|---|---|---|---|---|---|---|---|---|---|---|---|---|---|---|---|---|---|---|---|
| Pose Acc | 83.3 | 71.0 | 59.0 | 66.0 | 42.1 | 57.1 | 61.5 | 82.0 | 63.6 | 40.0 | 69.0 | 58.0 | 78.4 | 54.0 | 48.0 | 44.0 | 22.5 | 24.4 | 28.3 | 46.0 | 30.6 | 65.0 | 33.3 | 34.0 | 52.0 |
| 3D mIoU | 40.0 | 38.7 | 48.5 | 64.0 | 33.4 | 34.4 | 28.5 | 40.2 | 34.9 | 32.8 | 23.2 | 20.0 | 40.6 | 25.8 | 36.2 | 23.4 | 9.52 | 24.8 | 25.4 | 14.8 | 41.0 | 13.7 | 32.7 | 25.6 | 17.3 |
| CD ($10^{-2}$) | 0.37 | 2.51 | 7.53 | 3.03 | 1.27 | 1.44 | 9.09 | 0.35 | 5.42 | 17.3 | 2.42 | 8.51 | 3.02 | 16.1 | 1.29 | 6.43 | 10.7 | 16.6 | 4.34 | 2.65 | 4.84 | 0.74 | 10.1 | 6.09 | 23.5 |

| | | | | | | | | | | | | | | | | | | | | | | | | |
|---|---|---|---|---|---|---|---|---|---|---|---|---|---|---|---|---|---|---|---|---|---|---|---|---|
| Pose Acc | 50.0 | 92.0 | 65.0 | 44.2 | 64.2 | 20.0 | 68.5 | 36.7 | 48.9 | 65.7 | 71.0 | 39.0 | 73.0 | 42.2 | 65.7 | 52.5 | 60.0 | 34.0 | 95.0 | 32.0 | 50.0 | 38.0 | 57.0 | 53.0 | 44.0 |
| 3D mIoU | 31.0 | 47.0 | 28.1 | 31.9 | 58.7 | 14.7 | 54.3 | 30.3 | 31.0 | 56.0 | 34.8 | 32.6 | 24.9 | 36.4 | 38.1 | 24.3 | 37.0 | 28.8 | 29.7 | 30.1 | 24.0 | 44.5 | 42.4 | 18.6 | 27.6 |
| CD ($10^{-2}$) | 5.44 | 1.65 | 8.87 | 6.06 | 6.61 | 22.1 | 5.84 | 8.94 | 5.44 | 5.12 | 8.38 | 14.4 | 6.53 | 5.04 | 6.12 | 9.04 | 11.7 | 3.07 | 0.28 | 7.17 | 7.87 | 5.30 | 4.92 | 21.4 | 5.03 |

Table 1: **Quantitative results for 3D part detection from 2D images on 3D DST**. We use the pose accuracy, mean value of chamfer distance and mean 3D Bounding Box IoU over all parts to show that our method could precisely recognize 3D object parts from a single image.

## 5 EXPERIMENTS

To validate the effectiveness of Part321, we evaluate on diverse categories for 3D part detection in Section 5.2 and benchmark the quality of mesh-to-mesh correspondence in Section 5.3. We also evaluate Part321 on real image part datasets that only have 2D annotations to demonstrate that Part321's generalization ability on real images in Section 5.4. Furthermore, we demonstrate the effectiveness of important components of Part321 in Section 5.5 and introduce the computational cost in Section 5.6.

### 5.1 SETTINGS

**Training Details.** For each object category in 3D-DST (Ma et al., 2024), we want to use images generated from all annotated meshes (except the selected one) for testing and use the images generated from unannotated meshes for training. For categories that have few or no unannotated meshes, we will split $\frac{2}{3}$ of the total meshes for training (not using the part annotations).

**Metrics.** We use Chamfer Distance (CD) and 3D Bounding Box IOU to evaluate 3D part recognition. We compare the predicted 3D parts and ground truth parts after the camera and parts transformations to validate our model's ability to locate the parts. The pose estimation is regarded as accurate if the rotation error is smaller than $\frac{\pi}{6}$. For 2D segmentation, we use Mean Intersection over Union (mIoU) as the metric. Please note that these metrics, except for pose accuracy, are all conducted in a part-wise manner, and we report the mean value over all parts.

### 5.2 3D PART DETECTION

We conduct extensive experiments on DST-Part3D for 3D part detection from single 2D images. Since there are no directly comparable baselines, we construct a baseline by concatenating a state-of-the-art image-to-3D method (Long et al., 2024) and 3D part segmentation method (Yang et al., 2024) for comparison. We present results of 50 categories in Table 1.

As shown in Table 1, our method achieves promising results on diverse object categories, which proves that our learned deformable neural mesh effectively represents the categories and could

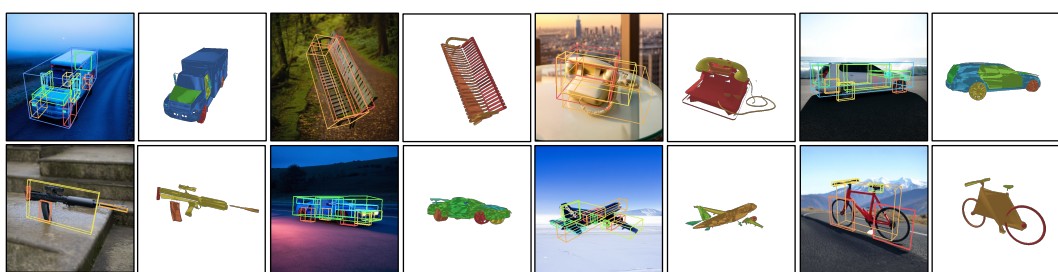

Figure 5: **Qualitative results on 3D DST.** The detected 3D bounding box and corresponding recognized parts show that our method could generalize to diverse object categories.

| Method | Pose Acc ↑ | CD $(10^{-2})$ ↓ | 3D mIOU ↑ |
|---|---|---|---|
| Wonder3D (2024) + SAMPart3D (2024) | - | 8.98 | 20.0 |
| Part321 | 53.5% | **7.16** | **32.5** |

Table 2: **Comparison with baselines on 3D part recognition.** Part321 outperforms models trained on the large-scale dataset by leveraging the learned deformable neural mesh.

| | | | | | | | | | | | | | | | | | | | | | | | | | |
|---|---|---|---|---|---|---|---|---|---|---|---|---|---|---|---|---|---|---|---|---|---|---|---|---|---|
| PartField (2025) | 28.9 | **70.9** | 40.9 | **97.2** | 59.4 | 29.4 | 20.4 | 27.5 | **52.4** | 30.6 | 20.3 | 32.6 | 32.6 | 39.0 | **65.8** | 39.0 | 35.6 | 27.2 | **46.6** | 20.4 | **40.7** | 19.5 | 36.3 | 44.3 | 10.5 |
| Capsule(ours) (2021) | **42.2** | 61.6 | **41.3** | 95.5 | **66.6** | **31.9** | **35.1** | **34.8** | 48.8 | **39.6** | **27.5** | **35.1** | **53.9** | **40.6** | 57.6 | **58.1** | **38.9** | **31.1** | 40.2 | **28.9** | 39.5 | **22.7** | **75.6** | **55.7** | **19.0** |

| | | | | | | | | | | | | | | | | | | | | | | | | | |
|---|---|---|---|---|---|---|---|---|---|---|---|---|---|---|---|---|---|---|---|---|---|---|---|---|---|
| PartField (2025) | 43.7 | 32.2 | **37.5** | 34.0 | 32.4 | 25.9 | 25.0 | **25.2** | 74.8 | 24.6 | 18.1 | **28.5** | 24.0 | 52.9 | 36.3 | **29.5** | **29.3** | **54.5** | 27.5 | **65.6** | 27.6 | **70.3** | 49.8 | 26.0 | 33.9 |
| Capsule(ours) (2021) | **43.7** | **41.6** | 35.0 | **49.4** | **39.1** | **42.6** | **35.5** | 20.1 | **77.3** | **30.5** | **31.2** | 27.6 | **24.2** | **59.8** | **45.4** | 23.8 | 23.9 | 38.5 | **29.6** | 64.2 | **33.9** | 51.0 | **60.4** | **35.3** | **42.9** |

Table 3: **Part transfer results on DST-Part3D.** We use the mean 3D Bounding Box IoU over all parts to benchmark the performance of mesh-to-mesh correspondence.

correctly locate the object parts in 3D space and deform the parts into suitable geometries. We do observe some categories propose more challenges due to very fine-grained annotations (e.g, steering wheel in golf cart), limited number of meshes (e.g., tram, rocket, and shopping cart), or huge shape variance across instances (e.g., kettle, mower).

Figure 5 visualizes the 3D bounding boxes of detected parts and their corresponding recognized 3D parts, where our method precisely detects the 3D pose and location of object parts and the predicted parts closely resemble the geometries in the image.

Table 2 shows the mean results across the 50 categories. Our framework outperforms the baseline significantly, which demonstrates the effectiveness of learning correspondences compared to large-scale training.

## 5.3 3D CORRESPONDENCE BENCHMARK

We evaluated two methods here: Capsule (Sun et al., 2021) and PartField (Liu et al., 2025). The 3D feature extractor of Capsule is self-supervisedly trained on 31,747 shapes across 13 categories from ShapeNet (Chang et al., 2015; Deprelle et al., 2019), whereas PartField's extractor is trained on 340k filtered shapes from Objaverse (Deitke et al., 2023) and 30k shapes from PartNet (Mo et al., 2019) with hierarchical part annotations.

Surprisingly, as shown in Table 3, Capsule(ours) shows a better performance in categories that have more fine-grained part annotations in DST-Part3D, especially when the part definition contains spatial information. Please refer to the Appendix A.2.2 for a qualitative comparison of the part transfer results. However, the quantitative results for both methods are not accurate enough, which can explain why Part321 degrades in several objection categories. We hope this benchmark can motivate works to incorporate more cross-shape supervision during training in the future.

| 2D mIOU ↑ | Police Car | Airliner | Bicycle | Jeep | Minibus | Mean |
|---|---|---|---|---|---|---|
| SegFormer (2021) | 39.57 | 37.25 | 23.21 | 37.02 | 32.82 | 33.97 |
| DeepLab v3+ (2018) | 44.54 | 35.71 | 23.74 | 34.47 | 31.60 | 34.01 |
| Matcher (20-shots) (2023b) | 32.85 | 22.31 | 29.95 | 30.63 | 35.29 | 30.21 |
| SLiMe (2023) | 37.60 | 35.23 | **38.34** | **46.78** | 32.61 | 38.11 |
| Part321 | **53.61** | **41.77** | 31.67 | 42.12 | **43.92** | **42.62** |

Table 4: **Quantitative results on real images.** Despite that Part321 performs the extra 3D recognition task, it outperforms 2D baselines with large-scale pretraining.

## 5.4 EVALUATION ON REAL IMAGES

To demonstrate that Part321 generalizes well to real images, we evaluate it on real image part datasets. We compared it with state-of-the-art 2D methods (Chen et al., 2018; Xie et al., 2021; Liu et al.,

Figure 6: **Qualitative results of 3D part recognition from our annotated real images.** We show the 2D segmentations and 3D parts rendered from two different views

|  | Pose Accuracy | CD $(10^{-2})\downarrow$ | 3D mIOU $\uparrow$ |
|---|---|---|---|
| w/o Scaling | **53.9** | 7.98 | 30.7 |
| w/o Constrain | 53.3 | 7.32 | 32.1 |
| w/o Deformation | 51.4 | 8.35 | 31.1 |
| Full Model | 53.5 | **7.16** | **32.5** |

Table 5: **Ablation study on 3D DST dataset.** We validate the necessity of our different components. The numbers are averaged across all 50 categories.

2023b; Khani et al., 2023) that are trained or fine-tuned on the same amount of synthetic images. The baselines include traditional segmentation methods (Chen et al., 2018; Xie et al., 2021) and also methods leveraging foundation models (Liu et al., 2023b; Khani et al., 2023).

Since the existing 2D part datasets do not correspond with the part definitions and fine-grained object categories in DST-Part3D, we select 279 images from the test and validation set of ImageNet (Krizhevsky et al., 2012) of five categories and annotate the part masks according to 3D annotations on the meshes. Table 4 and Figure 6 show that Part321 can generalize better to real-world images than 2D methods when trained only on synthetic images. For results on PartImageNet (He et al., 2022) and UDA Part (Liu et al., 2022), please refer to the Appendix A.2.3.

### 5.5 ABLATION STUDY

Table 5 shows the ablation study of important components in Part321. In the *w/o Scaling* setup, the object scale $S$ is set to be fixed during part optimization. For *w/o Constrain*, we remove the geometry consistency loss during optimization. The *w/o Deform* setting shows the results that no shape deformation is applied during inference. The results show that all the proposed components are essential to achieve the best 3D part detection performance.

### 5.6 COMPUTATIONAL RESOURCES

Our training takes about $10 \sim 48$ hours per category on a Titan RTX (depending on mesh count/size). Inference averages 1 minute per image for both 3D and 2D parts, while the baseline requires $\sim 12$ minutes (Wonder3D (Long et al., 2024): 7 min + SAMPart3D (Yang et al., 2024): 5 min).

## 6 CONCLUSION

To address the challenge of recognizing 3D object parts from 2D images. we firstly introduce DST-Part3D, that has $3,300$ annotated 3D parts on $475$ shapes from $50$ categories paired with $125,000$ realistic synthetic images. We then present Part321 to recognize 3D object parts from a 2D image using only one annotated mesh. Part321 establishes mesh-to-mesh and mesh-to-image correspondences to propagate part pseudo-labels across instances, allowing effective 3D part detector training with minimal supervision. Experiments demonstrate that Part321 outperforms 3D and 2D part detection tasks compared to alternatives. In addition, we use DST-Part3D to analyze the mesh-to-mesh correspondence obtained by different methods leveraging transferred 3d part labels, highlighting the key challenge in 3D part correspondence, which provides insight into future work.

# 7 STATEMENTS

## 7.1 REPRODUCIBILITY STATEMENT

We have made significant efforts to ensure the reproducibility of our results. The method details are described in Section 4. The training and testing settings are written in Section 5.1. More implementation details are provided in the Appendix A.1. We will release the source code as anonymous supplementary material. The DST-Part3D dataset will be also released publicly.

## 7.2 ETHICS STATEMENT

Our submission does not raise any questions regarding the Code of Ethics.

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

# A  APPENDIX

## A.1  METHOD DETAILS

### A.1.1  LEARNING FORMULATION OF MESH-TO-IMAGE CORRESPONDENCE

To relate the vertices in the deformable mesh with 2D images, we add neural features to its vertices and introduce the mesh-to-image correspondence, which is formulated as the similarity between features on each vertex $\theta_k$ and the features extracted $\Phi(I) = F \in \mathbb{R}^{c \times h \times w}$ from image $I$, where $\Phi$ is the feature extractor we need to train. We use the realistic synthetic images in DST-Part3D generated

by the meshes set $\{\mathfrak{M}_y\}$ for training. We only need to use the ground truth 3D pose and shape as the supervision, which requires no 3D part annotations.

To learn the correspondence, we use the mesh $\mathfrak{M}_y$ that is used to generate the image and calculate the world-to-screen transformation $\Omega$ using the known camera pose $Q$. To find 2D feature $f_k = F(p_k)$ at pixel $p_k$ that corresponds to the vertex $k$, we compute the projected location of each vertex on the feature map $p_k = \Omega(V_k)$. Besides, the visibility $o_k$ is determined for each vertex in the image, *i.e.*, $o_k = 1$ if vertex $k$ is visible, and vice versa.

Our learning objective is to enlarge the feature distance $\|f_k - \theta_l\|$ if $\|V_k - V_l\|$ is above a desired threshold, where $k$ and $l$ are indexes of two vertices. Such properties of features allow us to use differentiable rendering to find the optimal alignment of the vertices on the 3D model and corresponding locations on the 2D image. To achieve this, we use the contrastive loss (Bai et al., 2023; Wang et al., 2020) to learn the extractor:

$$\mathcal{L}_{\text{train}} = -\sum_k o_k \cdot \log\left(\frac{e^{\kappa f_k \cdot \theta_k}}{\sum_{v_l \notin \mathcal{N}_k} e^{\kappa f_k \cdot \theta_l} + \sum_{\beta_n \in \mathcal{B}} e^{\kappa f_k \cdot \beta_n}}\right), \tag{6}$$

where $\kappa$ is a preset softmax temperature, and $\mathcal{N}_k$ indicates the spatial neighborhood of $V_k$, which controls spatial error threshold of the correspondence. $\mathcal{B} = \{\beta_n \in \mathbb{R}^d\}_{n=1}^5$ is a set of background features that are pushed away from every vertex feature in the latent space to make the system robust to backgrounds.

At the same time, we adapt the vertex features $\theta_k$ using the momentum update strategy (Bai et al., 2023),

$$\theta_k \leftarrow o_k(1 - \sigma) \cdot f_k + (1 - o_k + \sigma \cdot o_k)\theta_k, \tag{7}$$

where $\sigma$ is the momentum for the update process.

### A.1.2 FEATURE INTERPOLATION PROCESS DETAILS

During the learning of mesh-to-mesh correspondence, we apply the feature interpolation process to compute the feature for every vertex on the meshes. Specifically, we train the PointNet++ (Qi et al., 2017b) encoder with sampled 1024 points $\{p_{y,i}\}$ from each mesh $y$, thus obtaining the features $\{\Gamma_{y,i}\}$ for those vertices. Then the features $\{\gamma_{y,k}\}$ on all vertices $\{V_k\}$ of the mesh $y$ are computed by the weighted sum of features of neighboring sampled vertices: $\gamma_{y,k} = \frac{1}{\sum_{j \in \mathcal{N}(k)} e^{w_{kj}}} \sum_{j \in \mathcal{N}} e^{w_{kj}} \Gamma_{y,j}$, where $\mathcal{N}$ denotes the neighboring vertices and $w_{kj}$ denotes the reciprocal of euclidean distance between vertex $k$ and vertex $j$.

### A.1.3 DEFORMATION NETWORK DETAILS

To train the deformation network, for each category, we use the annotated mesh as the template mesh, base on which the network predicts the 3D offsets of vertices given a shape latent. We use the training meshes in the category to train the network, which should deform the mesh into those meshes given the corresponding one-hot latent vectors. During inference, the deformed mesh could be seen as an interpolation among the selected meshes. Figure 7 shows that our deformation network could reshape the template into diverse geometries that resembles the object in the images.

### A.1.4 RENDERING VISIBILITY DETAILS

During Mesh-to-image correspondence training, we render the depth map $\mathbf{D} = Render(\mathfrak{N}_y, \Omega)$ and the vertex-to-camera distance $\mathbf{d}_k = \|Q - V_k\|_2$. Then the vertex visibility is computed as

$$o_k = \begin{cases} 0, \|\mathbf{D}_{p_k} - \mathbf{d}_k\|_2 > \tau_r \\ 1, \|\mathbf{D}_{p_k} - \mathbf{d}_k\|_2 \le \tau_r \end{cases}, \tag{8}$$

where $\tau_r$ is a preset threshold.

## A.2 MORE RESULTS

### A.2.1 MORE QUALITATIVE RESULTS ON 3D PART DECTION

As shown in Figure 8, we visualize the 3D bounding box and part prediction of more categories, which demonstrate our model's ability to generalize to diverse object shapes and part definitions. Please note that the first two predicted school bus parts are deformed from the same annotated mesh.

### A.2.2 QUALITATIVE COMPARISON OF PART TRANSFER

As shown in Figure 9, Capsule (Sun et al., 2021) produces more position-aware and less noisy part transfer results than PartField (Liu et al., 2025), despite being trained on only 13 categories with a relatively small amount of data.

### A.2.3 MORE RESULTS ON REAL IMAGE DATASETS

To further validate that Part321 can generalize well to real images, we compare with more 2D baselines on PartImageNet (He et al., 2022) dataset and UDA Part (Liu et al., 2022) as shown in Table 6 and Table 7. All methods are trained using only synthetic images and then evaluated on real images. Our method outperforms all 2D approaches despite that we perform the harder task of discovering 3D parts from 2D images. Note that we re-annotated the PartImageNet test images here to enable them to have more fine-grained annotations that correspond to our 3D part definitions (*e.g.*, for the car category, we separate 4 wheels and 2 doors with spatial information). As for the UDA Part, we merge the super fine-grained part definitions similarly.

Figure 10 shows the qualitative comparison between 2D baselines and Part321. Our method is more robust to the background and shows more precise part detection with spatial information (e.g, left engine and right engine).

| 2D mIOU ↑ | Car | Aeroplane | Bicycle | Mean |
|---|---|---|---|---|
| SegFormer (2021) | 28.85 | 39.68 | 26.72 | 37.47 |
| DeepLab v3+ (2018) | 28.73 | 38.69 | 34.76 | 37.23 |
| SLiMe (2023) | 27.09 | 20.35 | 31.32 | 26.25 |
| Part321 | **50.47** | **45.14** | **40.07** | **45.23** |

Table 6: Quantitative results of 2D segmentation on PartImageNet dataset show that Part321 outperforms baselines significantly when annotations are fine-grained (*e.g.*, Cars have 8 parts).

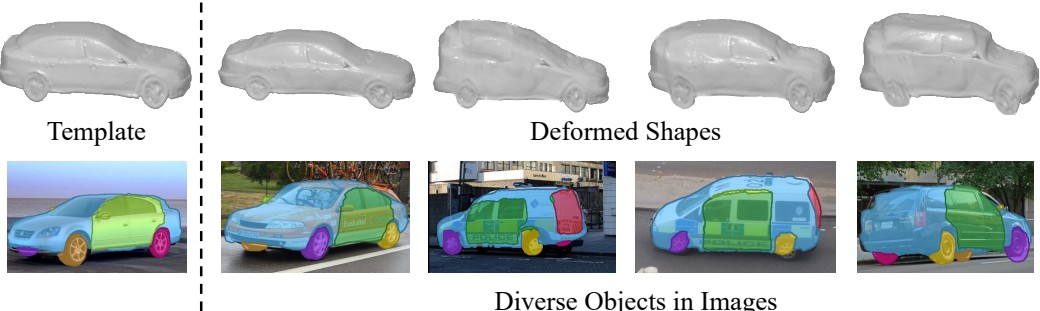

Template        Deformed Shapes

Diverse Objects in Images

Figure 7: Given a template shape and different shape latent, our deformation model could reshape the annotated mesh into diverse shapes to fit the objects in the images.

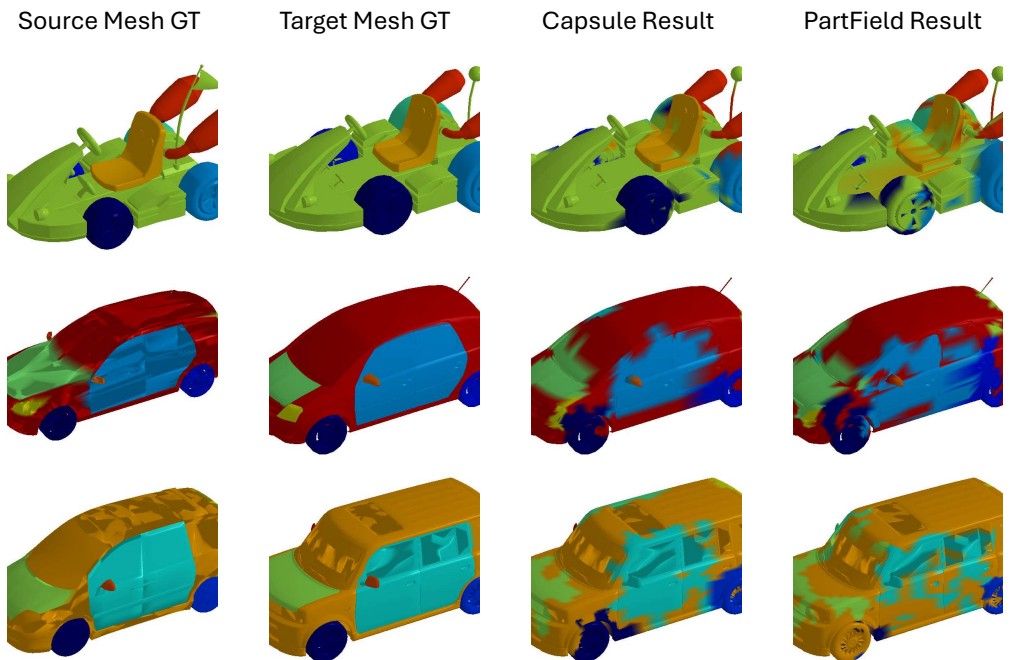

Figure 8: More qualitative results of 3D bounding box and 3D part prediction on diverse categories.

| Source Mesh GT | Target Mesh GT | Capsule Result | PartField Result |

Figure 9: Qualitative comparison of part transfer quality using the mesh-to-mesh correspondence.

### A.3 DST-PART3D DETAILS

#### A.3.1 PARTS TAXONOMY

Table 9 contains the parts taxonomy for the 50 rigid object classes in DST-Part3D.

#### A.3.2 DEMONSTRATION ON REALISM

To quantitatively demonstrate the realism of images in DST-Part3D can better bridge the domain gap, we compare the 2D part segmentation results with the synthetic image dataset of 3DCoMPaT++ (Slim et al., 2023). Synthetic images in 3DCoMPaT++ have diverse textures by compositing a wide range of high-quality materials on different parts, which is a good representative of direct rendering methods. Additionally, their synthetic images are rendered with a uniformly white background, which is also beneficial for evaluating the effectiveness of the realistic background generated by diffusion models.

In Table 8, we perform the comparison on the airplane category of 3DCoMPaT++. The diffusion-generated data following our pipeline uses the same 32 CAD models and viewpoints with 3DCoM-PaT++. "3DCoMPaT++(diffusion bg)" data is generated by replacing the foreground object of our diffusion-generated data with the foreground from 3DCoMPaT++. The evaluation is performed

Figure 10: Qualitative results of 2D part segmentation on PartImageNet dataset show that our prediction is more geometry-aware and is less affected by the complex background. Parts are represented by different color masks with highlighted boundaries.

| 2D mIOU ↑ | Car | Aeroplane | Bicycle | Mean |
|---|---|---|---|---|
| SegFormer (2021) | 24.33 | 35.92 | 38.30 | 32.85 |
| SLiMe (2023) | 25.45 | 28.82 | 40.98 | 31.75 |
| Part321 | **37.39** | **39.88** | **44.89** | **40.72** |

Table 7: Quantitative results for part segmentation on UDA Part show the robustness of our method with different part annotations. *w/ Pseudo* denotes the baseline trained with pseudo labeling.

on the PartImageNet (He et al., 2022) test set of airplanes. The synthetic-only setting is using SegFormer (Xie et al., 2021) and the UDA setting is using DAFormer (Hoyer et al., 2022), which adds the self-labeling modules to real images to the SegFormer.

As observed in the table, diffusion-generated object textures lead to better performance (*i.e.*, more realistic) than materials provided in 3DCoMPaT++ with improvements of 4.61 mIOU under the Syn-only setting and 4.79 mIoU under the UDA setting. The diffusion-generated background also benefits by comparing it to the white background, which is significant for observing that the background IoU improves from 85.40 to 92.33 under the UDA setting.

### A.3.3 ANNOTATION SCHEME

**(i) What parts to annotate per category:** One of the key challenges in annotating parts of 3D CAD models is the ambiguity of object part selection (*e.g.*, how to annotate the parts of a space shuttle). We divide our 50 rigid-object categories into five super-categories: car, airplane, bicycle, boat, and tool. We classify the super-category of each object category based on Wikidata and common knowledge. We then analyze what object parts are important in cognition and are tractable to be annotated in real images. Subsequently, we create part definition templates for each super-category, except tools, as the shapes of tools vary significantly. Therefore, we define part definitions individually for each tool category. Note that our part definitions are all recognizable from the object surface and we do not define parts that are internal structures. Annotators will check the shapes of their assigned CAD models first and then revise the provided part definitions only if necessary. **(ii) What principles to select part vertices:** We design several principles in selecting part vertices to guide the annotators to ensure high-quality and consistent 3D part annotations. Firstly, the annotated part vertex groups should be disjoint sets, and the union of all groups should contain every vertex in the original CAD models. Secondly, if a mesh face belongs to two connected parts, the annotator should not assign all three vertices to one part and should still assign the vertices based on where they are located. **(iii) Annotation quality inspection:** The annotation inspection is done by selected annotators whose annotations are high-quality during the annotation process, and the annotators will inspect the object categories that belong to the same super-categories of what they annotated in the annotation process.

### A.4 THE USE OF LARGE LANGUAGE MODELS(LLMS)

In paper writing, we use LLMs to polish writing. In addition, the image generation process of DST-Part3D requires LLMs to generate prompts.

| datasets | supervision | architecture | body | wing | engine | tail | bg | mIoU |
|---|---|---|---|---|---|---|---|---|
| 3DCoMPaT++ (white bg) (2023) | Syn-only | SegFormer (2021) | 25.45 | 11.28 | 7.64 | 5.64 | 83.43 | 26.69 |
| | UDA | DAFormer (2022) | 45.63 | 24.32 | 5.18 | 3.42 | 85.40 | 32.79 |
| 3DCoMPaT++ (diffusion bg) (2023) | Syn-only | SegFormer (2021) | 24.89 | 20.59 | 9.54 | 5.90 | 87.11 | 29.61 |
| | UDA | DAFormer (2022) | 44.76 | 25.62 | 6.79 | 3.81 | 92.33 | 34.66 |
| Diffusion generated (Ours) | Syn-only | SegFormer (2021) | 33.01 | 19.63 | **13.00** | **15.68** | 90.20 | 34.30 |
| | UDA | DAFormer (2022) | **56.43** | **26.37** | 10.71 | 9.61 | **94.13** | **39.45** |

Table 8: **Ablation on diffusion-generated textures and background**. "bg" is the abbreviation for background. "Diffusion generated" refers to synthetic data generated following our pipeline that uses the same 32 CAD models and viewpoints with 3DCoMPaT++. "3DCoMPaT++ (diffusion bg)" is generated by replacing the foreground object of diffusion-generated data with the foreground from 3DCoMPaT++. The results are evaluated on the PartImageNet test set of airplanes. Numbers are averaged over 3 random seeds.

| class ID | class name | parts taxonomy |
|---|---|---|
| n02690373 | airliner | engine†, fuselage, wing†, vertical_stabilizer, wheel(front, back_left, back_right), horizontal_stabilizer† |
| n02701002 | ambulance | wheel(front_left, front_right, back_left, back_right), door†, front_trunk, back_trunk, head_light†, frame, rearview† |
| n02749479 | gun | buttstock,magazine,barrel, gunbody |
| n02804414 | bassinet | stand, frame |
| n02814533 | beach wagon | wheel(front_left, front_right, back_left, back_right), door†, front_trunk, back_trunk, head_light†, frame, rearview† |
| n02835271 | recumbent bicycle | wheels(back, front), frame(paddle), handlebar, saddle |
| n02906734 | broom | handle, head |
| n02981792 | catamaran | sail, body |
| n03063689 | kettle | spout, body, handle |
| n03100240 | convertible | wheel(front_left, front_right, back_left, back_right), door†, front_trunk, back_trunk, head_light†, frame, rearview† |
| n03187595 | dial telephone | handset, dial, host, cord |
| n03272562 | tram | wheel†, door†, frame, rearview† |
| n03344393 | fireboat | top, body |
| n03345487 | fire engine | wheel†, door†, ladder_and_pump, frame, rearview† |
| n03417042 | garbage truck | wheel†, frame, front trunk, garbage_container |
| n03444034 | go-kart | wheel(front_left, front_right, back_left, back_right), frame, seat, engine |
| n03445924 | golfcart | wheel, frame, seat |
| n03481172 | hammer | handle, head |
| n03496892 | harvester | wheel†, frame, cutter, mirror |
| n03498962 | hatchet | handle, head |
| n03594945 | jeep | wheel(front_left, front_right, back_left, back_right), door†, frame, front_trunk, back_trunk, rearview† |
| n03599486 | jinrikisha | wheel(front, back_left, back_right), saddle, frame |
| n03642806 | laptop | keyboard, screen, body, touchpad |
| n03649909 | mower | wheel(front_left, front_right, back_left, back_right), steering_wheel, shaft, frame |
| n03670208 | limo | wheel(front_left, front_right, back_left, back_right), frame, rearview†, door†, head_light† |
| n03673027 | ocean liner | top, body |
| n03769881 | minibus | wheel(front_left, front_right, back_left, back_right), frame, door, rearview† |
| n03770679 | minivan | wheel(front_left, front_right, back_left, back_right), frame, door†, head_light† |
| n03785016 | moped | wheel(front, back), handlebar, frame, rearview |
| n03792782 | mountain bike | wheels(back, front), frame, handlebar, saddle |
| n03891251 | park bench | arm, backrest, beam, seat, leg |
| n03947888 | pirate ship | sail, body |
| n03977966 | police car | wheel(front_left, front_right, back_left, back_right), door†, front_trunk, back_trunk, frame, rearview† |
| n04037443 | race car | wheel(front_left, front_right, back_left, back_right), door†, front_trunk, back_trunk, head_light†, frame, rearview† |
| n04065272 | recreational vehicle | wheel(front_left, front_right, back_left, back_right), door†, front_trunk, back_trunk, head_light†, frame, rearview† |
| n04146614 | school bus | wheel(front_left, front_right, back_left, back_right), frame, head_light†, door†, rearview† |
| n04147183 | schooner | sail, bottom |
| n04204347 | shopping cart | wheel(front_left, front_right, back_left, back_right), basket, handle, frame |
| n04252225 | snowplow | wheel(front_left, front_right, back_left, back_right), frame, rearview†, cutter |
| n04266014 | rocket | engine†, fuselage, wing†, vertical_stabilizer, wheel†, horizontal_stabilizer† |
| n04285008 | sports car | wheel(front_left, front_right, back_left, back_right), door†, front_trunk, back_trunk, head_light†, frame, rearview† |
| n04465501 | tractor | wheel†, door†, arm_and_loader, frame, rearview† |
| n04467665 | trailer truck | wheel†, door†, front_trunk, trailer, head_light†, frame, rearview† |
| n04482393 | tricycle | wheels†, frame, handlebar, saddle, cargo_box |
| n04483307 | trimaran | sail, body |
| n04487081 | trolleybus | wheel(front_left, front_right, back_left, back_right), frame, door†, head_light† |
| n04507155 | umbrella | handle, canopy, frame |
| n04509417 | unicycle | wheels, frame, saddle |
| n04552348 | warplane | engine†, fuselage, wing†, vertical_stabilizer, wheel(middle, left, right), horizontal_stabilizer† |
| n04612504 | galley | sail, body |

Table 9: **Parts taxonomy of DST-Part3D.** †: indicates the left and right parts are separate part classes.

