# OpenReview forum: "A Dataset and Benchmark for 3D Part Recognition from 2D Images"
_ICLR.cc/2026/Conference — ICLR 2026 Conference Withdrawn Submission_

### Official Review · Reviewer_jsD6 · 2025-10-29

**Soundness:** 3
**Presentation:** 2
**Contribution:** 3
**Rating:** 4
**Confidence:** 3

**Summary:**

The paper introduces DST-Part3D, a dataset pairing 475 CAD shapes across 50 rigid categories with ~3,300 fine-grained 3D part annotations and 125,000 realistic synthetic images; it further proposes Part321, a one-shot 3D part recognition method from a single image that builds (i) mesh-to-mesh correspondence via vertex features and a deformation network, and (ii) mesh-to-image correspondence via contrastive, feature-level render-and-compare.

**Strengths:**

1. One-shot 3D part detection from a single image with explicit optimization of pose/scale/shape; equations & training details are complete.
2. Consistent quantitative gains over the Wonder3D→SAMPart3D baseline (3D mIoU 32.5 vs 20.0; CD 7.16 vs 8.98).
3. Practical reporting of training/inference costs (10–48 h/category; ~1 min image).
4. Real-image validation: on curated ImageNet subsets and PartImageNet/UDA-Part settings, the 2D projections outperform 2D baselines trained on synthetic data (Tables 4, 6, 7).

**Weaknesses:**

1. External generalization is thin. All core numbers are within the DST-Part3D ecosystem; no results on independent datasets (e.g., PartNet images, wild internet photos beyond the curated set). This limits claims of broad robustness.
2. Baselines are narrow and compound. The main comparison is a pipeline (Wonder3D→SAMPart3D) rather than category-level correspondences, such as NeMo/SHIC/MvDeCor adapted to parts. Including such baselines (or a stronger 3D-aware feed-forward alternative like PartField) would sharpen novelty.
3. Table 3 shows that both Capsule and PartField produce imperfect part transfers; the authors acknowledge this as a bottleneck, but there is no uncertainty estimate propagated to the inference stage (e.g., confidence-aware optimization).
4. Results note difficult categories (e.g., tiny parts, large intra-class variance), yet there is no quantitative per-factor analysis (occlusion, view angle, texture).
5. Presentation quality. Several typos reduce polish: “3D PART DECTION” (A.2.1), “objection categories” (p. 8), and encoding glitches (e.g., hyphenated “Ima￾geNet”), ect.

**Questions:**

1. Can you report results on independent real-image datasets (beyond your curated ImageNet split), or at least cross-category leave-out evaluations to quantify robustness?
2. How sensitive is inference to the initial pose sampling (144 initial poses) and to hyperparameters 𝑤_𝑐𝑜𝑛𝑠𝑖𝑠𝑡, 𝜏? Any ablation beyond Table 5?
3. Could you add confidence-aware correspondence (e.g., per-vertex reliability from the mesh↔mesh matcher) to guide optimization and reduce drift on tiny parts?
4. Please include direct baselines adapted from NeMo/SHIC/MvDeCor/PartField for part-level detection to position novelty more convincingly.
5. Runtime: your inference is ~1 min/image vs ~12 min baseline—what is the per-module breakdown and GPU type? (Table 5 reports only averages.)

---

### Official Review · Reviewer_hXDd · 2025-10-30

**Soundness:** 2
**Presentation:** 3
**Contribution:** 2
**Rating:** 6
**Confidence:** 3

**Summary:**

* The paper introduces DST-Part3D, a new 3D semantic part dataset that consists of 50 object categories, 475 shapes, approximately 3,300 annotated 3D parts, and 125,000 realistic synthetic images.
* In addition, it proposes Part321, an algorithm designed for one-shot 3D part recognition, where '3' denotes 3D part information, '2' denotes a single 2D image, and '1' denotes one annotated mesh. This framework enables 3D part recognition using only a single annotated mesh per category.
* The key contribution lies in establishing both mesh-to-mesh correspondence and mesh-to-image correspondence, allowing effective 3D part recognition with minimal supervision.
* Experimental results show that Part321 outperforms previous 2D segmentation and image-to-3D reconstruction approaches (e.g., Wonder3D + SAMPart3D). Moreover, despite being trained solely on synthetic data, the method generalizes well to real-image datasets such as PartImageNet and UDA Part.

**Strengths:**

* The proposed DST-Part3D overcomes the limitations of existing datasets such as ShapeNet-Part, PartNet, and 3DCoMPaT++, which rely solely on 3D inputs or simple synthetic renderings.
It combines realistic diffusion-generated images with fine-grained 3D semantic annotations, enabling training and evaluation of 3D part recognition from 2D images.
* The paper introduces a new “one-shot 3D part recognition” paradigm, demonstrating that category-level 3D part recognition can be achieved using only a single annotated mesh per category, effectively reducing supervision requirements.
* The authors conduct extensive experiments across both 3D metrics (pose accuracy, mean IoU, Chamfer Distance) and 2D metrics (mean IoU), along with a detailed ablation study that quantitatively validates the contributions of deformation, scaling, and geometry consistency losses.
* DST-Part3D serves as a new benchmark for evaluating 3D part correspondence methods, providing a platform to compare models such as PartField and Capsule through mesh-to-mesh part transfer tasks.
* Despite being trained purely on synthetic data, Part321 exhibits strong cross-domain generalization, outperforming 2D segmentation baselines such as SegFormer and DeepLab v3+ on real-image datasets, including PartImageNet and UDA Part.

**Weaknesses:**

* The experiments on real-image datasets are relatively small in scale (279 images across 5 categories), making it difficult to conclude that the domain gap between synthetic and real images has been fully addressed.
* Since the mesh-to-image correspondence learning relies heavily on contrastive learning, it may become unstable under view misalignment or occlusion conditions.
* While the diffusion-based synthetic data generation significantly improves realism, it may also introduce domain bias or style inconsistency across generated samples.
* The framework assumes “one annotated mesh per category”, which may limit generalization when there is large intra-category variation (e.g., trucks, furniture, boats).
* The DST-Part3D dataset includes only rigid objects, making it less applicable to articulated or deformable categories such as animals or humans.
* The paper provides limited qualitative or quantitative analysis of failure cases in part correspondence and lacks an in-depth discussion of the differences between articulated and rigid categories.
* The mesh deformation module requires per-vertex latent representations, which increases computational complexity and may limit scalability to high-resolution meshes.
* Minor Typos
  * Line 164: e.g. → e.g.,

**Questions:**

* How robust is the mesh-to-mesh correspondence when the part definitions become fine-grained or spatially asymmetric?

---

### Official Review · Reviewer_wZTB · 2025-10-31

**Soundness:** 3
**Presentation:** 3
**Contribution:** 3
**Rating:** 4
**Confidence:** 3

**Summary:**

This paper focuses on 3D part recognition from 2D images. To this end, a large-scale sythetic dataset named DST-Part3D is collected. This dataset consists of 125,000 images with 3D part annotations, which can be used to learn 3D part recognition, 2D part segmentation, and 3D part correspondence. Furthermore, this paper proposes Part321, which recognizes 3D parts from single images using one annotated mesh. This method pioneers one-shot 3D part detection by learning mesh-to-mesh and mesh-to-image correspondences. Experiments show that the proposed Part321 outperforms the competing methods on 3D part recognition, 3D part correspondence, and 2D segmentation.

**Strengths:**

1. The Part321 framework introduced in this paper are novel and interesting. To the best of my understanding, the proposed method—learning a category-level 3D prior representation and utilizing it to perform 2D/3D part segmentation by aligning with test images during inference—represents a relatively new paradigm that differs significantly from traditional feed-forward approaches.

2.  The constructed dataset DST-Part3D provides 125,000 realistic synthetic images with fine-grained 3D part annotations, which will contribute to the community of 3D part understanding.

3. The idea of learning mesh-to-mesh and mesh-to-image correspondences is well-motivated. The adopted learning objective (Eq.3, Eq.6) seems technically sound and reasonable.

4. The paper is clearly written and well-organized.

**Weaknesses:**

1. The competitive methods in this paper are somewhat limited. Experimental evaluations are conducted on three tasks: 3D part detection, 3D correspondence, and 2D segmentation. However, for the crucial 3D part recognition, only "Wonder3D + SAMPart3D" is introduced as a competing method for comparison (Table 2). More existing methods should be compared, especially "2D segmentation + 3D generation" pipelines. The performance comparison of 2D segmentation (Table 4) also lacks state-of-the-art segmentation methods like the SAM series. DeepLab v3+ and SegFormer are somewhat outdated.

2. The constructed DST-Part3D dataset is one of the main contributions of this work. Therefore, it is necessary to discuss and compare it with existing 3D part datasets and demonstrate the significance of the constructed dataset. In particular, the generation pipeline used is actually derived from another work, 3D-DST.

3. In Table 1, without quantitative results from other methods for comparison, it is difficult to evaluate the actual performance of the proposed method.

**Questions:**

Are there any applications that Part321 can achieve, which are not possible with existing 2D and 3D recognition methods (including combinations of these methods)?

---

### Official Review · Reviewer_7qeW · 2025-11-02

**Soundness:** 2
**Presentation:** 1
**Contribution:** 1
**Rating:** 2
**Confidence:** 4

**Summary:**

This paper introduces a novel dataset comprising 475 3D shapes paired with part annotation and corresponding 2D images, and proposes a one-shot part segmentation method applicable to both 2D and 3D inputs.

**Strengths:**

This paper addresses an important and interesting problem in 3D part segmentation, proposing a novel one-shot part segmentation method and introducing annotations for a small accompanying dataset.

**Weaknesses:**

1. **Dataset scale and positioning.** The proposed dataset is small in both scale (475 shapes) and diversity (50 object categories). The paper does not discuss or compare against prior 3D datasets with part annotations. It is unclear what makes this dataset unique, especially given existing 3D part datasets derived from ShapeNet. It is also unclear whether the limited scale and diversity are sufficient to support the paper’s claims and to enable meaningful follow-up work.

2. **Methodological limitations.** The proposed one-shot method has several inherent limitations:
   (a) It cannot handle arbitrary open-world categories; it requires a template mesh per category and category-wise training.
   (b) The deformation network does not support correspondences between meshes with different topologies (e.g., swivel chairs with four wheels vs. six wheels).
   (c) The provided 2D images are object-centric and free of occlusions. It is unclear how the method performs on more challenging, real-world images where objects are off-center and heavily occluded.

 3. **Clarity of exposition.** The paper is poorly written, with many details either missing or not presented in a self-contained manner. Several symbols are introduced without proper explanation. The overall presentation makes it difficult to follow the technical details. For example, Section 4.3 is not self-contained; despite considerable effort, it remains unclear how the components interact and what exactly is happening in that section.

4. **Experimental setup and terminology.** The experimental setup is insufficiently explained, and key terms are used inconsistently. Do “3D part detection” and “3D part recognition” refer to the same task? Likewise, do “part transfer” and “3D correspondence” denote the same objective? Please use consistent terminology and rigorously define each task, including inputs, outputs, and metric computations. Also specify how baseline methods are adapted to your evaluation protocol. For instance, for Wonder3D, how is object pose obtained? For PartField, how is it adapted to the one-shot setting? These details are currently missing.

5. **Choice of baselines.** The set of baselines appears outdated. It would be informative to evaluate whether recent vision–language models (e.g., Gemini, GPT-4/VLM variants) can understand object-part concepts from 2D images and how they compare under your tasks.

**Questions:**

Line 200: How is the shape latent obtained during training?

---

### Note · Authors · 2025-12-02

**Comment:**

We sincerely thank all reviewers for the thoughtful and constructive feedback. We appreciate the time and effort invested in evaluating our work and will carefully implement the suggested revisions to prepare a stronger version.

**Withdrawal Confirmation:**

I have read and agree with the venue's withdrawal policy on behalf of myself and my co-authors.